# ANTIBODY DESIGN WITH STEERABLE DISCRETE DIFFUSION

**Hugh Yeh**
Medical Scientist Training Program
Pritzker School of Molecular Engineering
University of Chicago
Chicago, IL 60637, USA
hughy@uchicago.edu

**Nikša Praljak**
Graduate Program in Biophysical Sciences
University of Chicago
Chicago, IL 60637, USA
niksapraljak1@uchicago.edu

**Zhijie Chen**
Department of Chemistry
University of Chicago
Chicago, IL 60637, USA
zhijiechen@uchicago.edu

**Thomas Walsh**
Pritzker School of Molecular Engineering
University of Chicago
Chicago, IL 60637, USA
tmwalsh@uchicago.edu

**Juan L. Mendoza**
Pritzker School of Molecular Engineering
Department of Biochemistry and Molecular Biology
Howard Hughes Medical Institute
University of Chicago
Chicago, IL 60637, USA
jlmendoza@uchicago.edu

**Jun Huang**
Pritzker School of Molecular Engineering
University of Chicago
Chicago, IL 60637, USA
huangjun@uchicago.edu

**Andrew L. Ferguson**
Pritzker School of Molecular Engineering
Department of Chemistry
University of Chicago
Chicago, IL 60637, USA
andrewferguson@uchicago.edu

## ABSTRACT

Antibodies are central to modern therapeutics, yet their discovery and optimization remain constrained by limited integration between computational design and empirical validation. Here, we present ADAPT, a generative antibody language model designed to operate in a seamless closed-loop cycle between *in silico* generation and high-throughput experimentation. ADAPT conditions antibody sequence generation on antigen context using an order-agnostic autoregressive discrete diffusion framework, enabling both localized complementarity-determining region (CDR) design and *de novo* full-length variable chain generation. We show that ADAPT learns biologically meaningful representations that capture global organization of antibody–antigen systems alongside local sequence organization within antibodies and antigens. Across computational benchmarks, ADAPT generalizes to antigens withheld during training. Experimentally, we validate ADAPT-generated nanobodies using yeast display. We further show that feedback can be used to steer generation toward specific functional properties, including improved humanness. Together, ADAPT provides a flexible and experimentally compatible framework for antibody design, enabling coordinated user defined optimization and rapid exploration of functional antibody sequence space.

# 1 INTRODUCTION

Antibodies and their engineered derivatives have transformed the treatment of cancer, infectious disease, and autoimmune disorders. Monoclonal antibodies targeting immune checkpoints, growth factor receptors, and viral antigens have become first-line therapies across a wide range of clinical settings (Chan et al., 2025). Notably, monoclonal antibodies such as the PD-1 inhibitors pembrolizumab and nivolumab have become standard first-line therapies for malignancies including non-small cell lung cancer, melanoma, and head and neck squamous cell carcinoma (Duan et al., 2020). Similarly, trastuzumab has drastically improved the prognosis of HER2-positive breast cancer (Swain et al., 2023). Beyond conventional immunoglobulins, alternative antibody formats such as bispecific antibodies, nanobodies, and single-chain variable fragments have further expanded the functional scope of antibody-based therapeutics, enabling applications ranging from drug conjugates to cell-based therapies (Carter & Rajpal, 2022; Chanier & Chames, 2019; Kalos et al., 2011). Central to this versatility is the ability of antibody variable domains to recognize molecular targets with high specificity. Despite their success, antibody efficacy is constrained by evolving disease-associated antigens. Viral escape mutations and tumor antigen remodeling drive the need for efficient discovery pipelines that can rapidly generate or adapt binders to shifting targets (Tuekprakhon et al., 2022; Chen et al., 2025; Zhang et al., 2020; Diamond et al., 2025).

Traditional antibody discovery approaches rely on immunization or large combinatorial display libraries followed by extensive experimental screening. Although powerful, these pipelines are inherently iterative and often optimize antibody properties in a serial manner. Improvements in one dimension, such as affinity maturation, can inadvertently degrade others, including stability, developability, or humanness, necessitating additional rounds of engineering (Carter & Rajpal, 2022). This sequential optimization paradigm increases experimental burden and can limit the ability to efficiently explore trade-offs among competing antibody properties.

Recent advances in machine learning have introduced new opportunities for antibody design. Sequence-based language models, such as AbLang Olsen et al. (2022), AntiBERTy (Ruffolo et al., 2021), IgLM (Shuai et al., 2023), LaMBO-2 (Gruver et al., 2023), and ProGen2-OAS (Nijkamp et al., 2023), benefit from the scale of available antibody sequence data (over a billion antibody sequences compared to ~5,000 resolved structures) and integrate naturally with high-throughput experimental assays, while structure-based approaches, such as Ig-VAE (Eguchi et al., 2022), can incorporate explicit information about antigen geometry to guide design. However, sequence-only models typically lack antigen context, limiting their ability to generate *de novo* antigen-specific antibodies. Conversely, structure-based models are constrained by the availability of high-quality structural data and iterative refinement is impractical due to structure input requirement. Hybrid models incorporating sequence and structure, such as AbDiffuser (Martinkus et al., 2023), DiffAb (Luo et al., 2022), RefineGNN (Jin et al., 2021)), while benefit from both sequence and structural information, often share the limitations of both as well. More broadly, most existing approaches are not designed to operate within an experimental feedback loop, restricting their capacity to systematically incorporate empirical measurements into successive rounds of design.

# 2 METHODS

To address key limitations of existing antibody design approaches, including lack of antigen-conditioned generation for language models, restricted design flexibility, and limited compatibility with iterative experimental feedback, we developed **A**ntibody **D**iffusion **A**nd **P**roperty **T**uning (**ADAPT**). ADAPT is a generative antibody language model that explicitly incorporates antigen sequences during training to learn antibody–antigen co-evolution and antigen specificity. To our knowledge, ADAPT is the first antibody design framework to combine antigen-aware generation, discrete diffusion–based CDR inpainting and full-chain synthesis, property steering, and allow continuous refinement across successive design–test cycles.

We adopt a sequence-based generative approach because it is naturally compatible with scalable experimental readouts such as yeast display coupled with next-generation sequencing (Fig.1A). While structure-based models provide valuable antigen geometry insights, structure-based feedback remains impractical at the scale required for iterative refinement, limiting opportunities for improvement. In contrast, ADAPT is designed to operate in a seamless closed-loop cycle between *in sil-*

Figure 1: **ADAPT is a closed-loop computation-experimental pipeline antibody discovery pipeline.** (A) Workflow to couple ADAPT designs with experimental feedback. (B) Illustration of order-agnostic autoregressive discrete diffusion.

*ico* generation and experimental validation, in which model-generated sequences are evaluated in high-throughput assays and iteratively reintegrated into subsequent rounds of fine-tuning allowing ADAPT to adapt in response to empirical outcomes. Further, ADAPT can be fine-tuned with conditional heads to steer generation toward specific targets and to simultaneously optimize multiple objectives, such as affinity, specificity, and humanness.

ADAPT is trained using an order-agnostic autoregressive discrete diffusion (Fig. 1B) first described by Hoogeboom et al. (2021), followed by expansion to proteins by Alamdari et al. (2023) and further expanded to allow for text conditioning by Praljak et al. (2024). We adopt and expand this training objective to be inclusive of multi-chain generation, allowing generation conditioned on global antigen-antibody context. Masked language models enable localized sequence editing, such as CDR inpainting, but are poorly suited for *de novo* full-length antibody generation (Wang et al., 2024). Conventional autoregressive protein language models generate sequences sequentially from the N to C terminus, committing early to framework residues that constrain downstream diversity(Wang et al., 2024). This limitation is particularly pronounced for antibodies, where specificity is dominated by CDRs, especially heavy chain CDR3 (HCDR3). ADAPT addresses both of these limitations. OA-ARDM allows ADAPT to support both CDR inpainting and full variable heavy and variable light chain (VH/VL) or nanobody (VHH) design. Inspired by the principles of somatic hypermutation, our discrete diffusion model iteratively refines antibody sequences in a stepwise denoising process (Fig. S1B). This enables the model to explore a large, rugged fitness landscape of all possible antibody sequences in a directed fashion, conditioned on antigen identity to identify a diverse pool of functional antibodies. See Supplementary A.3 for training details.

## 3 RESULTS

### 3.1 GLOBAL ANTIBODY-ANTIGEN CONTEXT AND LOCAL ORGANIZATION EMERGE IN ADAPT EMBEDDINGS WITHOUT DIRECT SUPERVISION

To assess whether ADAPT learned biologically meaningful representations, we examined the organization of its antibody embeddings. Because the model is jointly trained on antibody–antigen complexes, we were specifically interested in determining whether antigen-related information is reflected in antibody representations alone. To this end, embeddings were computed using only antibody sequences, without providing antigen inputs (see A.4 for details).

Visualization with t-SNE revealed that ADAPT embeddings organize antibodies in a hierarchical manner. At the coarsest level, the model separates single variable chains (VH/VHH or VL) from paired variable chains (VH/VL) (Fig. S2A). Within these broad categories, embeddings further stratify according to heavy- and light-chain species, germline heavy- and light-chain variable gene family, and light-chain type (Fig. 2B, 2C), S2C, S2D. This organization emerges without explicit gene annotations, indicating that the model learns family-specific framework and CDR features directly from sequence. Consistent with this, the embeddings also separate antibodies by species of origin, with prominent clusters corresponding to Camelidae, Hominidae, and Muridae (Fig. 2A). Notably, despite embeddings being computed from antibody sequences alone, local subclusters reflect antigen protein clusters (Fig. 2D and 2E). To compare the extent of antigen-related information captured by antibody embeddings, we computed the Calinski–Harabasz index using antigen cluster labels for embeddings from ADAPT and other antibody language models, revealing stronger antigen association in ADAPT representations. Specifically, in ADAPT's embeddings we saw clustering for distinct antigen and epitope groups such as HIV gp160, lysozyme c, spike glycoprotein, and volt-

Figure 2: **ADAPT embeddings capture global antibody-antigen and local antibody organization.** t-SNE of antibody embeddings labeled according to (A) heavy chain species, (B) germline heavy chain variable gene family, and (C) light chain type. (D) Calinski-Harabasz index on antibody embeddings with antigen cluster labels compared against other antibody language models. (E) t-SNE of antibody embeddings labeled according to antigen clusters.

age/pH gated potassium channels (Fig. 2E). This observation indicate that joint antibody–antigen training induces antibody representations that implicitly encode properties of the cognate antigen, even when antigen information is withheld at inference time. Together, the emergence of these separations without direct supervision suggest that ADAPT captures both global organization of antibody–antigen complexes and finer-grained organization within antibodies and antigens.

## 3.2 COMPUTATIONAL ASSESMENT OF CDR DESIGN AND DE NOVO ANTIBODY GENERATION

To evaluate ADAPT's generative performance, we considered two antibody design tasks of increasing difficulty: CDR inpainting and full *de novo* design of variable chains, in which complete VH/VL, VHH or VH sequences are generated (Fig. S1D).

For CDR inpainting, we evaluated ADAPT's ability to generate individual CDRs in both heavy and light chains, as well as all CDRs within a chain. Performance was assessed using amino acid recovery (Fig. 3A) and perplexity (Fig. S3A). To rigorously assess generalization, all evaluations were performed on a held-out SAbDab test set constructed by withholding entire clusters of antigens not observed during training (see A.1). To dissect the contributions of model pretraining and antigen conditioning, we compared ADAPT against ablated variants that ablated pOAS pretraining or without antigen/epitope information. Across all CDR regions, ADAPT with both pretraining and antigen conditioning significantly outperformed the model without pretraining (Wilcoxon signed-rank test; Fig. 3A, see S4 for parity plots). As expected, HCDR3 proved to be the most challenging region to generate, reflecting its high sequence variability. Notably, comparison to the antigen-ablated model revealed significant improvements in HCDR3 generation under both amino acid recovery and perplexity metrics (Fig. 3A, Fig. S3A), consistent with the central role of HCDR3 in determining antigen and epitope specificity (Xu & Davis, 2000). We further observed significant gains when generating entire heavy- or light-chain CDR sets, indicating that antigen information becomes increasingly important as larger portions of the antibody sequence are generated (Fig. 3A).

We next evaluated ADAPT on the more challenging task of full variable chain antibody design. Given the degeneracy of antibody sequence space, where multiple sequences can bind the same epitope, direct sequence comparison is insufficient. Instead, we used AlphaFold2-Multimer (AF2-M) predicted inter-chain TM-score (ipTM) as a proxy for binding likelihood (see A.6). All designs targeted antigens held-out from training. Because ipTM values are imperfect predictors of binding, we assessed each design relative to a reference antibody known to bind the same antigen. When tasked with designing full VH/VL, VHH or VH sequences, 77.3% of generated antibodies achieved ipTM values equal to or exceeding those of the corresponding reference antibodies, with strong correlation to reference ipTM scores (Pearson R = 0.814; Spearman $\rho$ = 0.846; Fig. 3B). Similar trends were seen for all CDR designs (Fig. S3B). These results indicate that ADAPT can generate antibodies with binding potential comparable to experimentally validated binders. Importantly, ipTM scores showed no significant correlation with sequence identity or similarity to the reference antibodies (Fig. S3C and S3D), suggesting that ADAPT's performance is not driven by memorization. Instead, ADAPT explores distinct regions of sequence space while preserving functional binding properties.

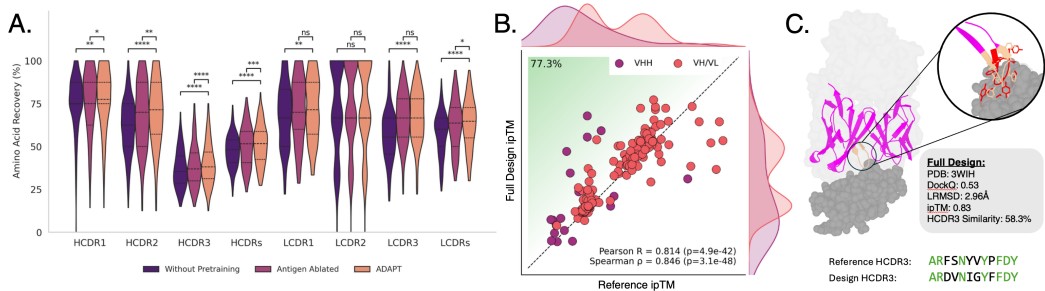

Figure 3: *In silico* **benchmark of ADAPT generation capabilities** (A) Amino acid recovery on inpainting tasks (B) AF2-M assessment of full chain design compared to reference VH/VL and VHH or VH (C) Example antibody generation for antigen in PDB 3WIH. Designed antibody (magenta) is shown in complex with the target antigen (gray), with zoomed views highlighting the HCDR3–epitope interface overlaying reference HCDR3 (red) with design HCDR3 (wheat). Design quality compared to reference antibodies is quantified using HCDR3 DockQ, and RMSD, along with HCDR3 sequence similarity. Beneath is a sequence alignment between reference HCDR3 and design HCDR3 with green residues indicating chemical equivalency.

To evaluate whether ADAPT designs targeted the intended epitope, we compared AF2-M–predicted antibody–antigen complexes with their corresponding reference crystal structures. Designed antibodies recapitulated the native docking orientation, achieving successful HCDR3 placement on the epitope as measured by DockQ ($\geq 0.23$). An example is shown in Fig. 3C and S5B, where the designed antibody adopts a binding pose consistent with the reference complex. Notably, despite modest HCDR3 sequence similarity (58.3%), the predicted HCDR3–epitope interface exhibited a low LRMSD (2.96 Å). Additional quantitative analyses of epitope targeting are provided in A.7.

### 3.3 EXPERIMENTAL VALIDATION AND STEERING DESIGN

We next sought to experimentally validate antibodies generated by ADAPT. As an initial test case, we focused on designing single-domain antibodies, VHH and VH, targeting the wild-type SARS-CoV-2 receptor-binding domain (RBD). To this end, we fine-tuned ADAPT on CoV-AbDab (Raybould et al., 2021), SARS-CoV-2-ADAPT, restricting training to antibodies and nanobodies targeting the RBD from wild-type virus as well as the Alpha, Beta, Delta, and Omicron BA.1/2/3 variants. Using SARS-CoV-2-ADAPT, we selected 100 designs against the wild-type RBD with the highest ipTM scores and subjected them to experimental testing using yeast surface display. Following two rounds of selection (see A.8, Fig. S6), we performed next-generation sequencing to quantify enrichment of individual designs relative to the naïve library. This analysis identified seven designs that were consistently enriched across both rounds of selection, as well as a total of ten designs enriched after the first round (Fig. 4A). Notably, we identified binders predicted to bind different neutralization epitopes (Fig. 4B). Further, we found that our functional designs exhibited sequence similarity as low as 74.6% to known sequences, especially the HCDR3 which exhibited sequence similarity as low as 54.5% (Fig. 4B).

To demonstrate the capacity of ADAPT's iterative design framework to generalize beyond training data, we evaluated performance on the held-out Omicron BA.4/5 RBD in an *in silico* directed evolution campaign. In each round, we generated 2,048 candidates targeting BA.4/5 and reintegrated sequences exceeding an ipTM threshold of 0.60 into subsequent training iterations. Across successive rounds, this closed-loop refinement led to increased ipTM of designs, indicating progressive improvement through iterative design and guidance despite the absence of these variants during initial training (Fig. 4C, see A.9 for more details). In an experimental directed evolution campaign, one would replace the ipTM oracle with experimental assays of antibody properties with which to fine-tune ADAPT.

To assess ADAPT's ability to steer generation toward specific properties, we trained ADAPT conditioned on SAbDAb species annotations (see A.3). By conditioning generation on species labels, we guided humanized and murinized antibody design, as assessed using BioPhi (Prihoda et al., 2022)

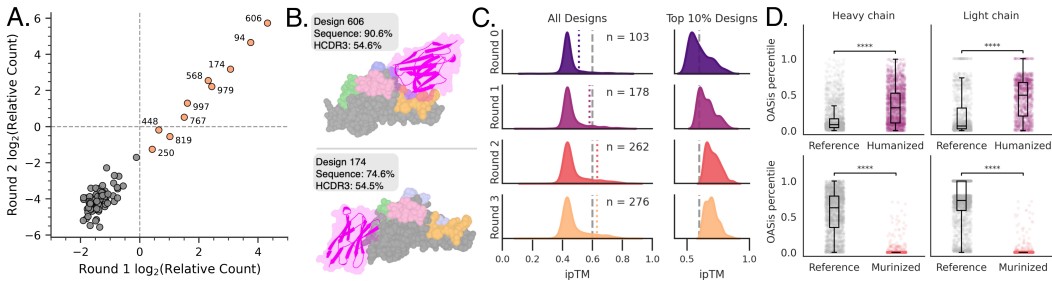

Figure 4: **Closed-loop experimental validation and guided antibody design with ADAPT.** (A) Relative read counts of nanobody designs post 2 rounds of selection. Highlighted are 10 relatively enriched designs out of the 100 tested designs. (B) AF2-M predicted structures of select designs. Designed nanobodies are magenta ribbons, and WT antigen SARS-CoV-2 spike RBD is shown as spheres with colors indicating the different neutralization epitopes. Nanobodies are labeled with overall sequence and HCDR3 similarity to known references. (C) KDE plot of iteratively generated ipTM-guided antibodies against held-out Omicron BA.4/5. Gray dashed line indicate ipTM of 0.60 and colored dashed lines indicate top 10 percentile of designs. To the right of each round is a zoomed in distribution of top 10 percentile. (D) Humanization and murinization measured by OASis percentile (1 = human, 0 = non-human) separated by heavy and light chain.

as an external oracle (Fig. 4D; see A.10 for more details). These results demonstrate that ADAPT can incorporate conditioning signals to guide antibody design toward desired properties.

## 4    CONCLUSION

In summary, we present ADAPT, an experimentally validated closed-loop framework for antigen- and property-guided antibody design. ADAPT supports both targeted CDR redesign and *de novo* full-length antibody generation, and we demonstrate its capabilities through computational analyses and experimental validation of nanobodies targeting the SARS-CoV-2 RBD. By integrating empirical feedback into iterative design cycles, ADAPT enables refinement and steering of antibody generation toward desired functional properties. Ongoing work includes expanded experimental characterization of designed antibodies, iterative design driven directly by experimental binding data, generalization to unseen or sparsely represented antigens, and coordinated multi-objective guidance across affinity and developability objectives.

## 5    ACKNOWLEDGMENTS

We are grateful for the support of the University of Chicago's Research Computing Center for assistance with the calculations carried out in this work. This research also used resources of the Argonne Leadership Computing Facility, which is a U.S. Department of Energy Office of Science User Facility operated under contract DE-AC02-06CH11357.

MEANINGFULNESS STATEMENT

ADAPT explores how meaningful biological representations can emerge by modeling antibody–antigen co-evolution from sequence alone. The model learns antibody representations that reflect biology of the immune system by implicitly organizing antibodies by germline family, species, and chain type. By conditioning on antigen context, these representations shift in response to selective pressure from evolving antigens and can be further steered using experimental feedback and clinically desired properties. Together, this work demonstrates how learned representations encode evolutionary coupling between antibodies and antigens while supporting *de novo* antibody design and prediction in ever-changing and evolving biological systems.

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

## A  APPENDIX

### A.1  DATASET PROCESSING

Paired Observed Antibody Space (pOAS; 1.9M sequences) (Kovaltsuk et al., 2018) database sequences were filtered using ANARCI (Dunbar & Deane, 2016) to remove non-recognizable entries. ANARCI was further used to trim sequences to variable domains and to assign region annotations using the Chothia numbering scheme (Al-Lazikani et al., 1997), including identification of CDRs. Sequences were clustered based on CDR sequence similarity using MMseqs2 (Steinegger & Söding, 2017) easy-cluster with the following parameters: $s = 7.5, min - seq - id = 0.90, c = 0.9, cov - mode = 1, cluster - mode = 1$. Data were split by CDR clusters into training (98%), validation (1%), and held-out test (1%) sets to prevent sequence-level leakage.

Structural Antibody Database (SAbDab; approx. 4500 complexes) (Dunbar et al., 2014) entries were filtered to retain only samples containing a single antigen. Single-chain variable fragment (scFv) entries were separated into heavy and light chains using ANARCI, and all sequences were trimmed to variable domains. Region annotations for VH/VL, VHH, and VH sequences were assigned using ANARCI with the Chothia numbering scheme. Antibody epitopes were defined as the C-$\alpha$ of antigen residues within 12 Å of C-$\alpha$ of antibody residues. Antigens were clustered using MMseqs2 easy-cluster with parameters: $s = 7.5, min - seq - id = 0.4, c = 0.8, cov - mode = 1, cluster - mode = 1$. The dataset was split into training, validation, and test sets based on antigen clusters, with approximately 90% of data used for training, 5% for validation, and 5% held out for testing. This cluster-based split ensures that antigens in the test set are distinct from those seen during training, enabling assessment of generalization to unseen targets.

The Coronavirus Antibody Database (CoV-AbDab) (Raybould et al., 2021) was filtered to retain antibodies with reported binding to the SARS-CoV-2 receptor-binding domain (RBD). Antibodies targeting the wild-type virus and the Alpha, Beta, Delta, and Omicron BA.1/2/3 variants were used for model training and validation, using a random split (X sequences for training and Y for validation). Antibodies capable of binding Omicron BA.4/5 variants were held out in their entirety to assess generalization. All sequences were trimmed to VH/VL, VHH or VH variable domains and annotated using ANARCI with the Chothia numbering scheme.

### A.2  ADAPT ARCHITECTURE

ADAPT is built on an efficient linear-attention transformer architecture. The base model contains approximately 340M parameters and comprises four sequential transformer blocks, each with an internal depth of 16 attention–feedforward layers and a hidden dimension of 512.

Input sequences are first encoded at the amino acid level. Residues are tokenized using a dictionary of 27 tokens, consisting of the 20 standard amino acids, an unknown amino acid token, a start token, an end token, two chain separator tokens, a pad token, and an absorbing (masked) state used during diffusion. Amino acid tokens are mapped to learned embeddings and serve as the primary sequence representation. In addition to amino acid identity, regional information is encoded through a separate region tokenization scheme. Regions are tokenized using a dictionary of 20 tokens, corresponding to four framework regions and CDRs for the light chain, four framework regions and three CDRs for the heavy chain, two chain separator tokens, start and end tokens, two tokens to represent the epitope and antigen (Fig. S1C). Region tokens are embedded and passed through an MLP to produce region-specific representations. Positional and diffusion step time are additionally incorporated. Diffusion time is encoded using a sinusoidal time embedding, while positional information along the sequence is represented using an axial positional embedding. The region embeddings are fused with the axial positional embeddings to form the final positional representation used by the transformer.

The maximum sequence length is 1280 tokens, comprising up to 149 tokens for the heavy-chain variable region, 149 tokens for the light-chain variable region, and 978 tokens for the antigen, with the remaining four tokens reserved for start, end, and chain separator tokens. All sequences begin with a start token and terminate with an end token, with chain separator tokens used to delineate heavy and light chains as well as antibody–antigen pairings.

For conditioned variants of ADAPT, a lightweight conditioner MLP produces block-specific conditioning embeddings, which are injected after each transformer block. For our species informed ADAPT, we utilize SAbDab labels of heavy and light chain species to label each sample as either from human, mouse, Camelidae family, unknown or missing.

## A.3 TRAINING ADAPT

ADAPT pretraining is outlined in Figure S1A. ADAPT is first pretrained on paired antibody sequences from pOAS. Although pOAS lacks antigen annotations, it provides a rich corpus for learning fundamental principles of antibody sequence organization, including heavy–light chain pairing and framework variability. The model is subsequently fine-tuned on the SAbDab, enabling it to learn antigen-conditioned generation and epitope targeting. Following this stage, ADAPT can be further refined through targeted fine-tuning on individual antigens or on clinical and biophysical objectives, focusing generation toward specific regions of sequence space and enriching for properties such as affinity or humanness. During training, each example consists of the heavy chain, light chain, and antigen sequences concatenated with chain-specific delimiter tokens. To further support context-aware generation, the model receives region-level annotations identifying framework and complementarity-determining regions (H/LCDR1–3), as well as antigen epitope labels (Fig. S1C). These annotations allow ADAPT to learn the internal local organization of antibody sequences and their global interactions within antibody–antigen complexes, reinforcing biologically meaningful structure without relying on explicit geometric conditioning.

$$\mathcal{L} = \mathbb{E}_{t\sim\mathcal{U}(1,\ldots,D)}\mathbb{E}_{\sigma\sim\mathcal{U}(S_D)}\left\{\frac{1}{D-t+1}\sum_{k\in\sigma(\geq t)}\log p(x_k \mid x_{\sigma(<t)})\right\} \tag{1}$$

We train ADAPT using an order-agnostic autoregressive discrete diffusion objective, following prior autoregressive diffusion formulations (Hoogeboom et al., 2021). Given an input sequence complex $x \in \mathbb{R}^D$, we sample a prefix length $t \sim \mathcal{U}(1,\ldots,D)$ and a random permutation $\sigma$ over sequence complex positions. The model conditions on the observed prefix $x_{\sigma(<t)}$ and predicts the remaining positions $x_{\sigma(\geq t)}$. To encourage learning of antibody-antigen coevolution, permutations are sampled uniformly over positions spanning both antibody and antigen sequences, ensuring that residues from both component appear in the conditioning context or prediction target. The loss averages the log-likelihood $\log p(x_k \mid x_{\sigma(<t)})$ over all unobserved positions $k$, normalized by $D - t + 1$, ensuring uniform contribution across different prefix lengths. Taking expectations over both $t$ and $\sigma$ encourages consistent predictions across arbitrary generation orders.

$$\mathcal{L} = \mathbb{E}_{t\sim\mathcal{U}(1,\ldots,D)}\mathbb{E}_{\sigma\sim\mathcal{U}(S_D)}\left\{\frac{1}{D-t+1}\sum_{k\in\sigma(\geq t)}\log p(x_k \mid x_{\sigma(<t)}, z_c)\right\} \tag{2}$$

We extend this objective to a conditionally guided order-agnostic autoregressive discrete diffuion loss by incorporating an auxiliary conditioning variable $z_c$ into the model likelihood. The training procedure remains identical, sampling a prefix length $t \sim \mathcal{U}(1,\ldots,D)$ and a random permutation $\sigma$ over sequence positions. The model predicts the unobserved positions $x_{\sigma(\geq t)}$ conditioned on both the observed prefix $x_{\sigma(<t)}$ and the conditioning label $z_c$, which may encode clinical or biophysical properties associated with the sequence. As before, permutations are sampled uniformly over positions spanning both antibody and antigen sequences, encouraging learning of coevolutionary structure, while conditioning on $z_c$ enables the model to learn associations between sequence patterns and desired properties and to steer generation toward specific design objectives.

## A.4 ANTIBODY REPRESENTATIONS

To compute antibody sequence representations, we extracted hidden states from the final transformer block of ADAPT at the last diffusion timestep $t = L$, corresponding to the fully denoised sequence. For a given antibody sequence of length L, token-level embeddings were mean-pooled across sequence positions to obtain a fixed-dimensional representation. Special tokens, including start, end, and chain-separator tokens, were excluded from this aggregation.

To assess whether antibody representations encode antigen-related information, embeddings were computed using antibody sequences alone. Antigen sequences and epitope annotations were withheld during representation calculation, ensuring that any observed antigen-associated structure arises from representations learned during joint training rather than from explicit antigen input at inference time. Representations were visualized with t-SNE (Maaten & Hinton, 2008).

## A.5 ANTIBODY GENERATION

ADAPT supports two sequence generation modes: targeted inpainting and full-length generation. In inpainting mode, user-specified residues or regions are replaced with a absorbing state (mask) token, and the diffusion timestep $t$ is set to the corresponding noise level to control the extent of sequence redesign. In full-generation mode, the model generates complete chains with only the antigen sequence.

ADAPT additionally accepts template constraints that specify antibody format and regional structure. These inputs include chain configuration (VH/VL or VHH/VH), total sequence length, and region-level constraints such as CDR boundaries and HCDR3 length, which are provided through region annotation input channels. Together, these controls enable flexible design of antibodies ranging from localized CDR edits to constrained or unconstrained *de novo* sequence generation.

For computational benchmarks of CDR inpainting and full variable chain design (Fig. 3 and Fig. S3), 10 designs were generated per target, and the top-performing design was reported. For epitope targeting experiments (Fig. S3), the highest-confidence AF2-M prediction was selected, defined as the complex with the top ipTM score among five AF2-M predictions generated from the 10 designs for each antigen.

One-sided Wilcoxon signed-rank test was used to assess improvement in ablation study. Statistical significance thresholds are indicated by asterisks, with * corresponding to $p < 0.05$, ** to $p < 0.01$, *** to $p < 0.001$, and **** to $p < 0.0001$. P-values were adjusted for multiple hypothesis testing using the Benjamini–Hochberg false discovery rate (BH-FDR) procedure, applied separately within each region.

## A.6 ALPHAFOLD2-MULTIMER METHODS

To evaluate full-length and all-CDR antibody designs in Fig. 3B and Fig. S3B, we used AlphaFold2-Multimer (AF2-M) (Evans et al., 2021) as implemented in ColabFold v1.5.5 (Mirdita et al., 2022). For these benchmarking analyses, PDB templates were not provided in order to avoid information leakage or bias when comparing model-generated sequences to reference antibodies. Reported performance in Fig. 3B and Fig. S2 reflects the mean predicted inter-chain TM-score (ipTM) across AF2-M model predictions. For ranking candidate designs prior to experimental validation, AF2-M was instead run with structural templates enabled to improve prediction accuracy. In this setting, designs were ranked using the maximum ipTM score across all model predictions. All AF2-M runs were performed with a maximum of five recycling iterations.

## A.7 EPITOPE TARGETTING

Specific epitope targeting is a central challenge in antibody design. To evaluate epitope targeting performance, we assessed three design modes on a held-out test set: full variable chain design, full CDR design, and HCDR3-only design. Because HCDR3 is the primary determinant of antigen specificity ](Xu & Davis, 2000), we quantified targeting accuracy using DockQ (Mirabello & Wallner, 2024), measuring alignment between the designed HCDR3–epitope interface and the reference crystal structure.

To control for potential structural priors in AF2-M, which may partially infer epitopes from previously observed antigen-antibody complexes with similar antigens, we compared design DockQ scores against shuffled baselines. For the full-chain baseline, antibodies binding unrelated antigens were randomly paired with each target. For CDR shuffling, CDR loops were sampled from antibodies recognizing different antigens, and for the HCDR3 baseline, loops were similarly drawn from non-cognate binders. As expected, HCDR3 design achieved the highest success rate (DockQ $\geq 0.23$). Across all design modes, we observed more correctly targeted epitopes relative to the

shuffled baseline (Fig. S3A). Representative examples of successful epitope targeting across design modes are shown in Fig. S3B.

To ensure reliable evaluation, we restricted analysis to antibody–antigen pairs for which AF2-M accurately recovered the reference epitope when folding the reference sequences (Fig. S5C). This filtering yielded 44 evaluable test cases (171 complexes were DockQ-compatible, with one excluded due to lack of contact). For each case, we estimated the improvement over the shuffled baseline by bootstrapping the difference in DockQ scores 10,000 times (Fig. S5D).

HCDR3 design showed the strongest performance, with nearly all designs (>99%) exceeding their shuffled counterparts. Full-chain design also demonstrated substantial gains, with approx. 93% of designs improving upon baseline. In contrast, CDR-only design exhibited the lowest rate of improvement. We hypothesize that this reflects a constrained optimization regime: unlike full-chain design, which can compensate for suboptimal interactions through distributed mutations, CDR-only design limits structural adaptability, while simultaneously providing fewer positional constraints than HCDR3-focused generation. As a result, errors may be harder to recover, leading to reduced targeting accuracy.

### A.8    Yeast Display, Selection, and Deep Sequencing

To experimentally assess the binding behavior of ADAPT-designed antibodies, we performed yeast display followed by magnetic-activated cell sorting (MACS) and flow cytometric analysis (Fig. S6A). Antibody libraries generated by ADAPT were first amplified by PCR and transformed into Saccharomyces cerevisiae, where variants were displayed on the cell surface via the Aga1p/Aga2p system (Boder & Wittrup, 1997). Displayed antibodies were epitope-tagged to enable monitoring of surface expression and were stained with fluorescently labeled antigen for binding assessment.

Following transformation, yeast libraries were subjected to two rounds of MACS to enrich for antigen-binding clones. Antigen-stained cells were incubated with magnetic beads and passed through magnetic columns to isolate antigen binding populations. Flow cytometry was used to quantify the fraction of antigen-binding cells before selection and after each MACS round. As shown in Fig. S6B, successive rounds of MACS resulted in enrichment of antigen-binding yeast relative to the naïve library. Sequencing read counts for each antibody design are obtained using next-generation deep sequencing of the library following each selection round. We define the $\log_2$ enrichment, $\log_2 \text{RE}(\text{round}_N)$ (Eq. 3), as the base-2 logarithm of the ratio between sequencing read counts $C_{\text{round}_N}$ observed after selection round $N$ and the corresponding read counts $C_{\text{naive}}$ in the naïve library, providing a measure of relative enrichment during selection.

$$\log_2 \text{RE}(\text{round}_N) = \log_2 \left( \frac{C_{\text{round}_N}}{C_{\text{naive}}} \right) \tag{3}$$

To further evaluate apparent binding affinity, enriched library after two rounds of MACS was analyzed by flow cytometry across a range of antigen concentrations (10, 5, and 2.5 µM), alongside a negative control lacking antigen (Fig. S6C). Antigen-positive populations were readily detectable at higher concentrations, and remained relatively unchanged at lower concentrations, indicating affinity under 2.5 µM. There is ongoing work to experimentally measure the binding affinity of our top designs.

### A.9    Iterative Training and Design

To design antibodies against Omicron BA.4/5 in the absence of variant-specific training data, we implemented an iterative design framework initialized with templates from antibodies known to bind earlier SARS-CoV-2 variants. These templates were paired with the Omicron BA.4/5 receptor-binding domain (RBD) to perform full variable chain design.

Starting from SARS-CoV-2-ADAPT (round 0), we generated 2,048 candidate sequences. Complex structures were predicted with AF2-M, and designs with a maximum ipTM score $\geq 0.60$ were retained. The selected sequences were then incorporated into the training set to fine-tune the model for the subsequent round. For each new round, templates were resampled from the updated training distribution, additional designs were generated, and the same selection procedure was applied.

This design–evaluation–fine-tuning cycle was repeated for three rounds. Each successive round of refinement led to an increase in number of successful designs as determined by an ipTM score $\geq$ 0.60.

## A.10 GENERATION WITH SPECIES GUIDANCE

We condition and fine-tune ADAPT following the same procedure described in A.2, A.3 and A.9. Antibodies are labeled at the chain level as belonging to human, mouse, the Camelidae family, or as missing species annotation. To evaluate ADAPT's ability to guide generation by species, we perform both humanized and murinized guided designs against antigens in SAbDab. For humanization, we select a non-human subset of SAbDab and condition generation on human species labels to produce full variable chain designs. For murinization, we select the human subset of SAbDab and condition generation on mouse species labels to produce full variable chain designs. In both cases, the corresponding SAbDab subsets serve as the reference sequences for evaluation. We quantify humanization (for humanized designs) and de-humanization (for murinized designs) using the BioPhi (Prihoda et al., 2022) OASis percentile score, where a percentile score of 1 indicates highly human-like sequences and 0 indicates non-human sequences. BioPhi was used with Chothia numbering and CDR definitions. In cases where BioPhi was unable to annotate sequences, a default humanness score of 0 was given. Statistical significance is assessed using a one-sided Wilcoxon signed-rank test. Statistical significance thresholds are indicated by asterisks, with * corresponding to $p < 0.05$, ** to $p < 0.01$, *** to $p < 0.001$, and **** to $p < 0.0001$.

## B SUPPLEMENTARY FIGURES

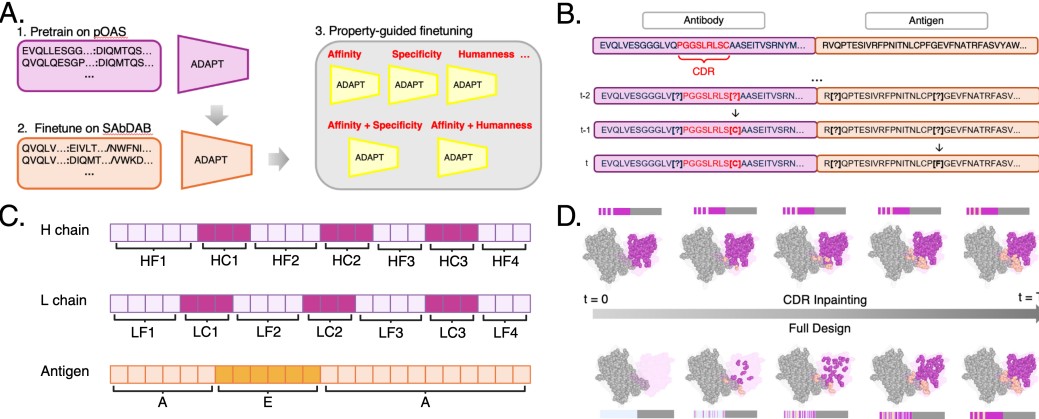

Figure S1: **Training protocol and design modes for ADAPT**. (A) Pretraining on paired antibody sequences from pOAS followed by antigen-conditioned fine-tuning on SAbDab. (B) Illustration of the order-agnostic autoregressive discrete diffusion process over antibody and antigen sequences. (C) Region-level annotations for antibody and antigen sequences. Antibody regions are labeled using ANARCI and tokenized per residue, where H and L denote heavy and light chains, F denotes framework regions, C denotes complementarity-determining regions (CDRs), and numeric indices specify individual framework and CDR segments. Antigen residues are labeled as epitope (E) or non-epitope antigen (A). (D) Inference modes supported by ADAPT, including CDR inpainting and full-chain antibody design.

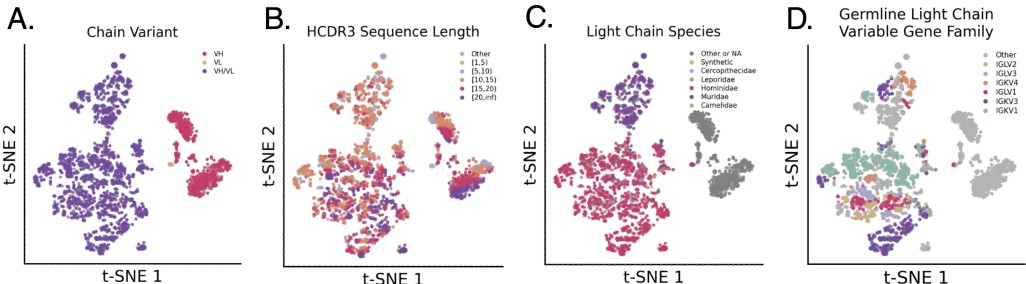

Figure S2: **Organization of antibody sequence representations learned by ADAPT.** t-SNE projections of antibody embeddings colored by different biological attributes. (A) Chain variant, distinguishing VH-only, VL-only, and paired VH/VL antibodies. (B) HCDR3 sequence length, grouped into length bins. (C) Light chain species origin. (D) Germline light chain variable gene family. Across panels, ADAPT embeddings exhibit structured organization that reflects known biological properties without explicit supervision.

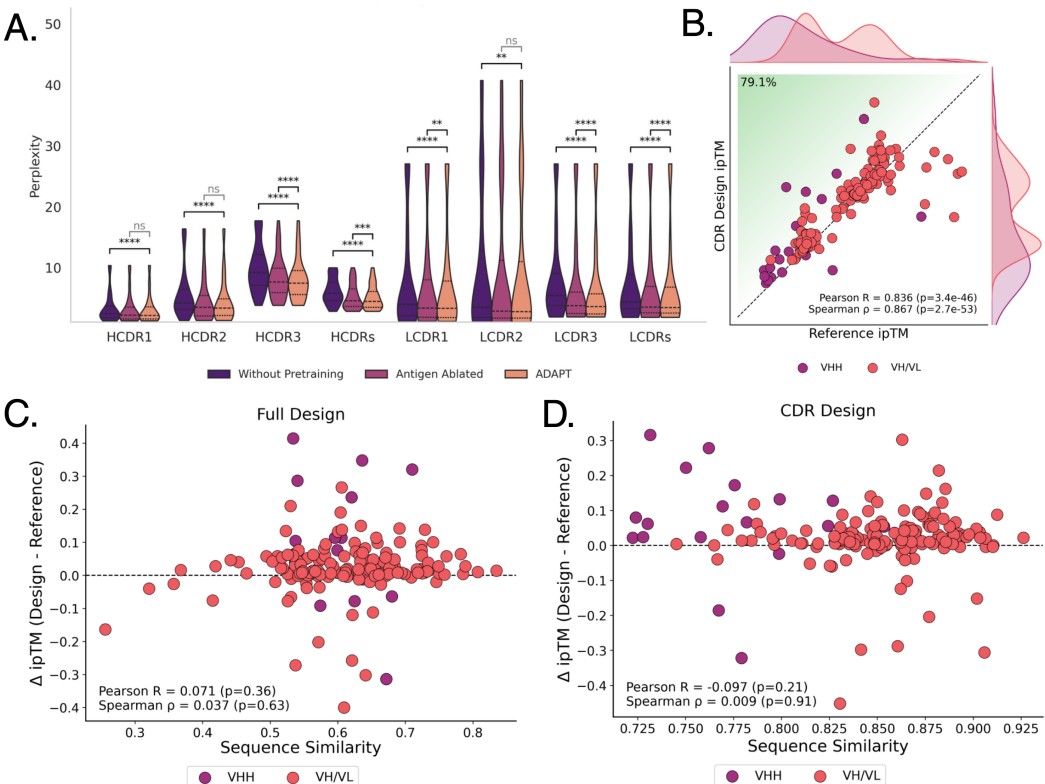

Figure S3: **Additional assessment of ADAPT generation capabilities.** (A) Perplexity of generated residues across antibody complementarity-determining regions (CDRs), comparing models without pretraining, with antigen information ablated, and full ADAPT. Lower perplexity indicates improved sequence modeling. (B) AF2-Multimer assessment of all CDR inpainted design compared to reference VH/VL, VHH, and VH. Relationship between sequence similarity to nearest train sequence and change in ipTM for (C) full-chain designs and (D) all CDR designs relative to reference antibodies.

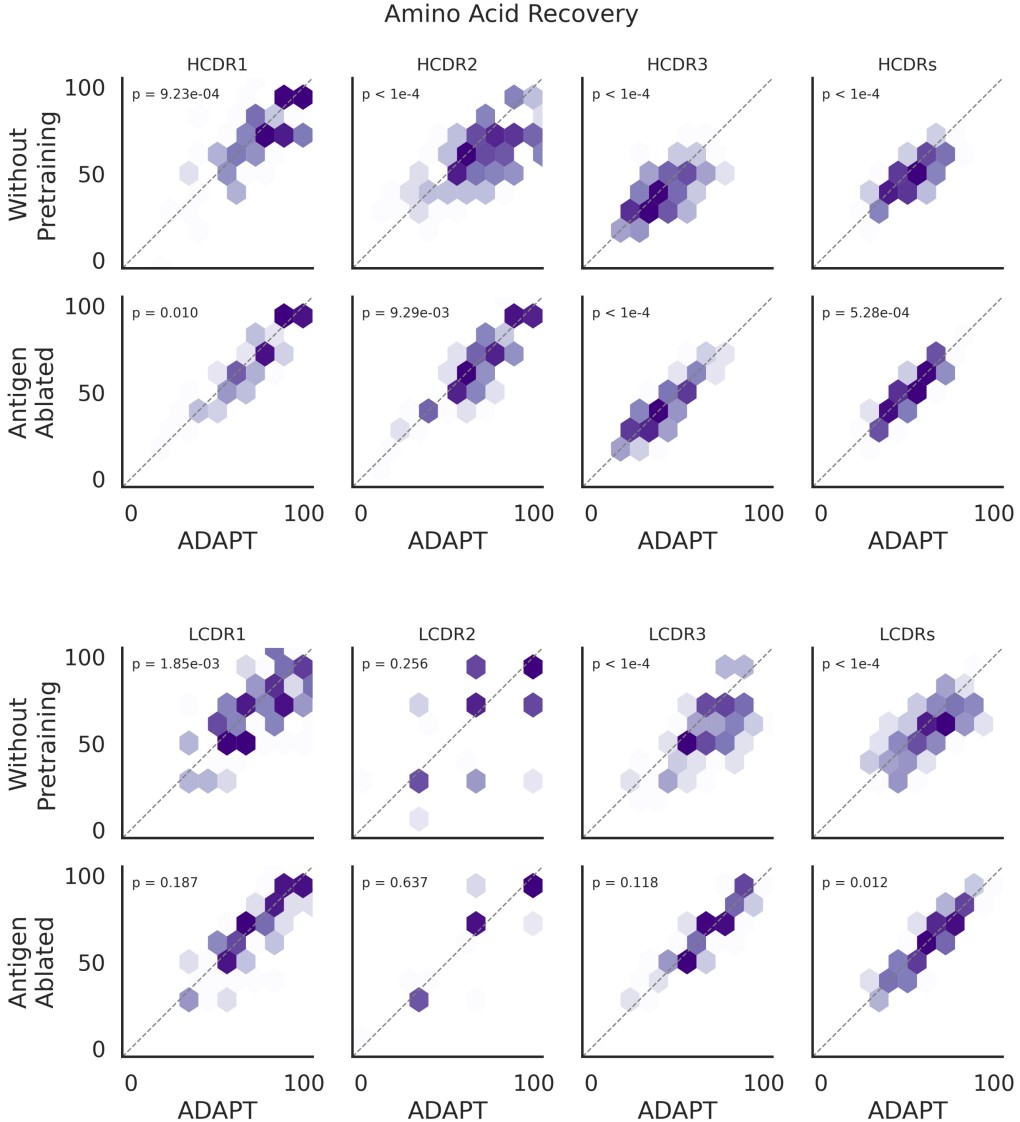

Figure S4: **Amino acid recovery parity plots comparing ADAPT to ablated models.** Hexbin density plots show per region amino acid recovery rates for heavy- and light-chain CDRs. The x-axis denotes ADAPT performance, while the y-axis shows performance of ablated variants (without pretraining or without antigen conditioning). Darker hexagons indicate higher point density. Points below the diagonal indicate datasets where ADAPT outperforms the ablation, whereas points above indicate cases where the ablated model performs better. P-values correspond to paired Wilcoxon signed-rank tests across regions.

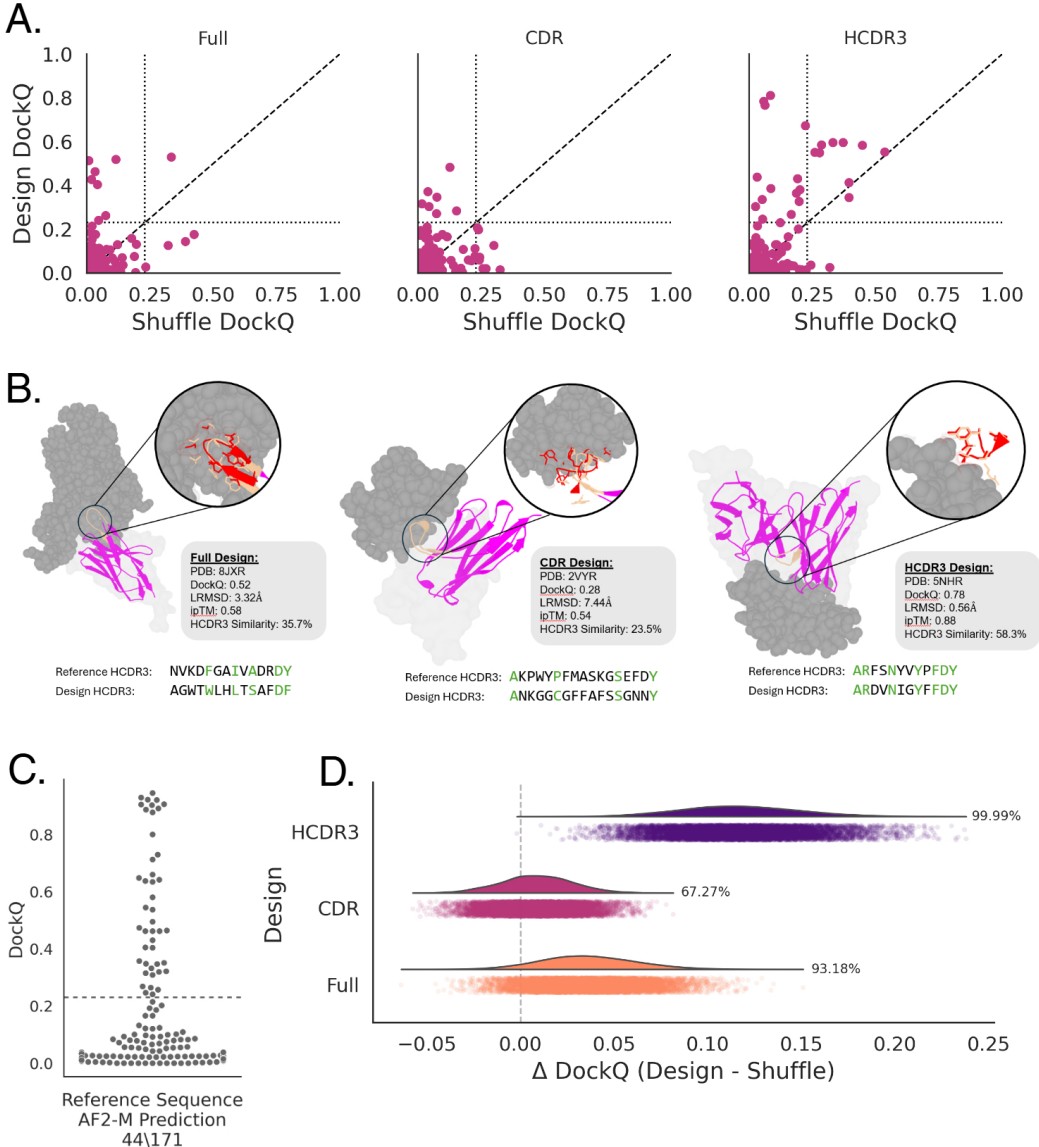

Figure S5: **Epitope targeting assessment of ADAPT-designed antibodies under different design modes.** (A) Comparison of DockQ scores for designed antibodies versus sequence-shuffled controls (see A.7) across three design modes: full-variable region, CDR-only, and HCDR3-only. Each point represents a predicted complex, with dashed lines indicating parity and dotted lines marking the DockQ success threshold (0.23). (B) Example of antibodies generated by ADAPT using full variable region design (left), CDR-only design (middle) and HCDR3 design (right). Designed antibodies (magenta) are shown in complex with the target antigen (gray), with zoomed views highlighting the HCDR3–epitope interface overlaying reference HCDR3 (red) with design HCDR3 (wheat). Structural similarity to reference antibodies is quantified using HCDR3 DockQ, and RMSD, along with HCDR3 sequence similarity. Beneath is a sequence alignment between reference HCDR3 and design HCDR3 with green residues indicating chemical equivalency. (C) DockQ distribution for AlphaFold-Multimer predictions of held-out reference sequences. (D) Distribution of $\Delta$DockQ (design - shuffle) across design modes. Most designs yield positive gains, with HCDR3-only design showing the largest and most consistent improvements. Percentages indicate the fraction of designs exceeding shuffled controls. See A.7 for additional details.

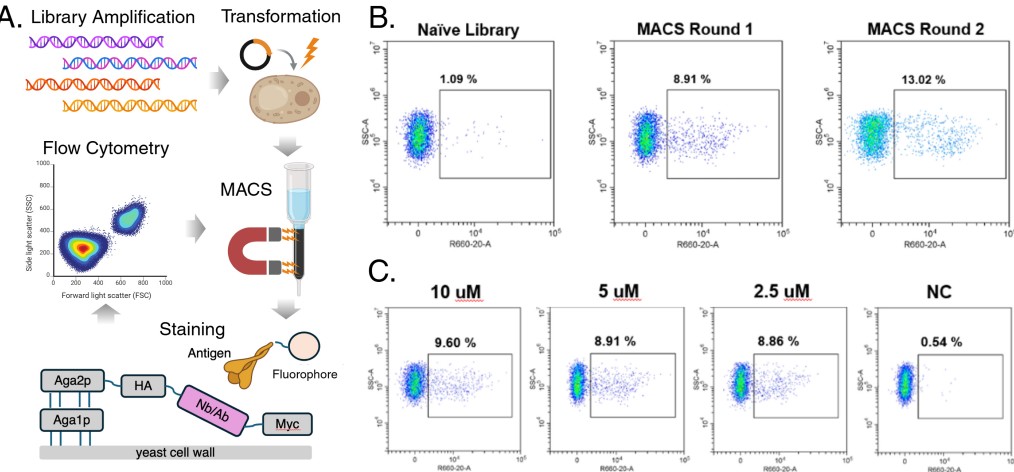

Figure S6: **Yeast display–based experimental validation of ADAPT-designed antibodies.** (A) Schematic of the yeast display workflow, including library amplification, yeast transformation, antigen staining, magnetic-activated cell sorting (MACS), and flow cytometric analysis. Antibody variants are displayed on the yeast surface via Aga1p/Aga2p and detected using fluorescently labeled antigen. (B) Flow cytometry plots showing enrichment of antigen-binding yeast populations across successive magnetic activated cell sorting rounds, starting from the naïve library. Percentages indicate the fraction of antigen binding cells within the gated population. (C) Antigen titration after 2 rounds of MACS with a negative control (NC) suggesting affinity is below 2.5 $\mu$M affinity.

