# OpenReview forum: "Antibody design with steerable discrete diffusion"
_ICLR.cc/2026/Workshop/LMRL — ICLR 2026 Workshop LMRL Poster_

### Official Review · Reviewer_MSpB · 2026-02-20

**Rating:** 7
**Confidence:** 5

**Review:**

The authors train an any-order-autoregressive model (equivalent to a masking diffusion model) on paired OAS and SAbDab.

They show their model's embeddings correlate with species and V, D, and J types (a trivial result). They show they also correlate with the actual antigen of antibodies in SAbDab, but this observation could just as likely be confounding as meaningful structure.

Fig 3 shows that they fit the training distribution. They show that their fit is statistically significantly better when including antigen information; however the magnitude of the impact is infinitesimal and could also be explained by confounding in SAbDab.

Finally they showed that samples from their model conditioned on a protein from SARS-CoV-2 generates some Ab that can bind the protein. Yet there are no baselines in silicon or in vitro -- I wonder if unconditionally generated sequences could do well. I also wonder if there is data leakage from the fact that there are sequences in SAbDab that bind this protein.

I don't see this modeling paradigm leading to interesting results in the future. Yet this paper is an earnest attempt by the authors, and I think will spark interesting discussion in the workshop, so I recommend accept.

---

### Official Review · Reviewer_YpQg · 2026-02-23
**ADAPT: Antigen-Conditioned Order-Agnostic Discrete Diffusion for Closed-Loop Antibody Design**

**Rating:** 7
**Confidence:** 3

**Review:**

## Summary
This paper introduces **ADAPT**, an antigen-aware generative antibody language model trained with **order-agnostic autoregressive discrete diffusion** to support both **CDR inpainting** and **full VH/VL (or VHH) generation**, and demonstrates a closed-loop design concept with **yeast display** validation and **property steering** (e.g., humanness).

### Quality
- The methodology is grounded in a clear diffusion-style likelihood objective (Eq. 1–2) and provides concrete implementation details (tokenization, max length, conditioning injection).
- The computational evaluation uses held-out **antigen-cluster splits**, reducing trivial antigen leakage and making generalization claims more credible.

### Clarity
- The paper clearly motivates why OA-ARDM is preferable to standard AR LMs for antibodies (avoid early framework commitment; enable inpainting and full generation).
- Some key training/finetuning hyperparameters are deferred to supplement sections, so reproduction still likely requires code release.

### Originality
- The main contribution is a **sequence-only, antigen-conditioned** diffusion LM designed explicitly for **iterative experimental feedback loops**, combining generation + steering in one framework.
- Conceptually, the modeling ingredients (OA-ARDM + conditioning) are adapted from prior work, so novelty is more in **integration + antibody-specific design modes** than in a fundamentally new generative paradigm.

### Significance
- Experimental yeast display enrichment of multiple designed nanobodies is a strong step beyond purely in silico studies and supports practical relevance.
- The “closed-loop” claim is partly aspirational since some iterative refinement is demonstrated using an **ipTM oracle**, not direct experimental measurements yet.

## Pros
- **Antigen-conditioned generalization:** Evaluation explicitly withholds antigen clusters during training and still shows improved inpainting and full design performance.
- **Flexible design modes:** Supports HCDR3/CDR inpainting and full-chain generation within one OA-ARDM formulation.
- **Experimental validation:** Yeast display + NGS enrichment provides tangible evidence of binder discovery potential.
- **Property steering:** Conditioning on species labels demonstrates controllable humanization/murinization with an external humanness oracle.

## Cons
- **Oracle dependence:** Many key results (ipTM, DockQ from AF2-M structures) rely on AF2-M as an evaluation/selection oracle, which can miscalibrate binding and epitope targeting.
- **Limited wet-lab depth:** Yeast display shows enrichment and coarse affinity bounds (suggested <2.5 µM), but lacks quantitative kinetics/affinity for top hits.
- **Benchmark protocol choice:** Reporting the *best-of-10* design per target may inflate apparent success rates versus average-case generation quality.
- **Steering scope:** Property guidance is shown for humanness, but multi-objective steering (affinity/specificity/developability simultaneously) is not yet convincingly demonstrated beyond framing.

---

### Meta-Review · Area_Chair_eFsY · 2026-02-27

**Recommendation:** Accept (Poster)
**Confidence:** 5

**Metareview:**

Accept.

---

### Decision · Program_Chairs · 2026-03-02

**Decision:**

Accept (Poster)

**Comment:**

Please see the meta-review.